# Domestic Factors as Determinant of Sickness Absence with Psychiatric Disorders: A Scoping Review of Nordic Research Published between 2010–2019

**DOI:** 10.3390/ijerph20136292

**Published:** 2023-07-04

**Authors:** Gunnel Hensing, Varsha Rajagopalan, Carin Staland-Nyman

**Affiliations:** 1School of Public Health and Community Medicine, Sahlgrenska Academy at University of Gothenburg, 405 30 Gothenburg, Sweden; varsha.nr07@gmail.com (V.R.); carin.staland_nyman@hh.se (C.S.-N.); 2School of Health and Welfare, Halmstad University, 301 18 Halmstad, Sweden

**Keywords:** domestic work, sickness absence, psychiatric disorders, gender equality

## Abstract

Uneven division of domestic factors may contribute to sex differences in sickness absence with psychiatric disorders. The aim of this scoping review was to compile current Nordic research on domestic factors and sickness absence with psychiatric disorders. A systematic search was performed to identify studies from the Nordic countries published between 1 January 2010 and 31 December 2019. Twelve studies were included. Marital status, family situation, work-home interference (in both directions), social affiliation, and loss of child/young adult (suicide, accident, or natural death) were identified as measures of domestic factors. In 8 of the 12 studies, domestic factors were used as co-variates, while four used them as the main exposure. Social affiliation, home-to-work conflict, and total workload were not associated with the outcome. One study found that parents with children older than two years, widows/widowers, and those divorced or unmarried had an increased risk of sickness absence with psychiatric disorders. One study found that work-to-home conflict was associated with sickness absence with stress-related diagnoses in men, and with other mental disorders in women. Finally, one study found that losing a son or daughter aged 16–24 years increased the risk of future sickness absence with a psychiatric disorder regardless of the cause of death. Future studies need to develop concepts, study designs, and measurements to move this research area further. In particular, the concept of “unpaid domestic work” needs theoretical and empirical development.

## 1. Introduction

Women outnumber men in sickness absence with a psychiatric disorder [1]. The observed differences have been discussed in relation to the possible influence of women’s total workload with high labor participation rate and responsibility for the unpaid work in household and family [2,3]. It is previously known that a burdensome responsibility for household and family work, including an unequal division of such work between genders, may be associated with decreased mental health [4,5,6,7]. Moreover, in a global systematic review of the associations between unpaid work and mental health (including both mental health symptoms and mental health diagnoses certified by physicians), unpaid work was associated with poorer health in women, while the association was less evident in men. An unequal division of unpaid/domestic work between genders was also discussed as exposing women to greater risk of poorer mental health [8].

In relation to sickness absence with psychiatric disorders, a French prospective study including individuals simultaneously exposed to high levels of demands from paid and domestic work showed an elevated risk of sickness absence with psychiatric disorders, with women showing higher rates than men [9]. The adjusted relative risk (RR) for sickness absence with depression was 6.58 (95% CI 3.46–12.50) in women and 3.55 (1.62–7.77) in men. The gender distribution into male vs. female-dominated work sectors might influence sickness absence due to differential work-related exposures [10,11]. However, in a study using a fixed effect model, comparisons were made between women and men in the same occupations and/or workplaces [12]. Differences were reduced but still a third remained unexplained. The unexplained part raises the question of whether non-work-related factors play a role in sickness absence with psychiatric disorders.

Sickness absence with psychiatric disorders is a challenge for many societies, social insurance systems, and employers [13,14,15]. Mental health problems cost countries in the EU over 600 billion Euros per year, of which absenteeism and presenteeism are the largest parts [16]. In many cases, sick leave is needed and relevant. However, there is a higher risk of recurrence, an increased risk of long-term sickness absence, or even a risk of permanent marginalization from the labor market in sickness absence with psychiatric disorders [17,18,19,20]. Thus, from the perspective of societies, employers, and individuals affected, better knowledge of factors outside paid work that might be contributing to sickness absence with psychiatric disorders is needed. The aim of this scoping review was, therefore, to compile and describe current Nordic research on domestic factors and sickness absence with psychiatric disorders. With this scoping review, we add to existing knowledge of domestic factors and their impact on mental health. Today, sustainable health is likely to be achieved if both factors in paid work and domestic factors are better understood as detrimental to and promotive of health. 

The Nordic region, including Denmark, Finland, Iceland, Norway, and Sweden, was chosen only to ease the comparison of the literature acknowledging the similarities in the welfare systems and availability of register data. 

More specifically, the research questions were:(i)What number and type of studies have been performed in the Nordic countries and published over the years 2010–2019 regarding the significance of domestic factors for sickness absence with psychiatric disorders?(ii)What type of measures of domestic factors have been used?(iii)What associations have been found between domestic factors and sickness absence with psychiatric disorders?(iv)What knowledge gaps can be identified for future research?

## 2. Materials and Methods

The study was designed as a scoping review which allows researchers to chart the underpinning concepts of a research area and review existing evidence and sources [21,22]. Based on the key objectives and the broad focus of the review, the methodology prescribed by Arksey et al. (2003) for a scoping review was adopted. The review consisted of five steps: scoping, searching, screening, data extraction, and data analysis. Reporting of the methods and findings was guided by the Preferred Reporting Items for Systematic Reviews and Meta-Analysis (PRISMA) criteria. 

### 2.1. Search Strategy

The search for scientific articles on 1 January 2010–31 December 2019 was carried out in the databases Pubmed, PsycInfo, and Sociological Abstracts. In addition, a literature search was carried out in the Gender Studies Database (GSD), LIBRIS, and Women’s Research. LIBRIS is a national search service providing information on titles held by Swedish universities and research libraries and twenty public libraries. Hand searches were carried out in the following identified relevant gender journals: Journal of Gender Studies (Sweden), Journal of Gender Research (Norway), Women, Gender & Research (Denmark), and Gender Research (Finland). The literature search was supported by information specialists from the library at the University of Gothenburg. Surveys were conducted by using combinations of keywords and formulated search blocks (see Appendix A). A total of 1475 studies were initially identified, which was reduced to 1167 publications after a duplicate cleanup. A detailed selection process including a relevance review is presented in the flowchart in Figure 1.

### 2.2. Study Selection

The checklist for the inclusion and exclusion criteria can be found in Appendix B. 

The assessment of relevance based on the title and abstract (*n* = 1167) was performed independently in Rayyan by two researchers. (One was the first author GH. The second was PhD Maria Boström who has quit working in academia and declined to be a co-author.) These two researchers also assessed the relevance of 214 full-text papers. When the two relevance assessments were carried out, 57 articles remained. The second author (VR) scrutinized these articles for exposure measures of domestic factors. For this part, an explorative approach was undertaken to understand which variables could be defined as measures of domestic factors. Initially, an inclusive approach was used, and possible variables were listed. These variables were discussed and included if they concerned domestic work, which in this study included both unpaid work and the broader aspect of family life. Twelve studies were finally included. 

### 2.3. Charting Data

Data from the selected publications (*n* = 12) were charted by GH and VR to record the author, date of publication, study design, study population, main exposures, and confounders used in the studies. When faced with disagreements on narrowing down the themes, we discussed our reasoning and reached a consensus.

### 2.4. Collating, Summarizing, and Reporting Results

The data were summarized by frequency, type of publication, study population, and the type of data used. Each publication was coded based on its relevance to these identified categories. Therefore, one study could have more than one type of exposure measure associated with it. Finally, gaps in the literature were identified. 

## 3. Results

### 3.1. Characteristics of Included Literature

All included studies had an epidemiological study design; the most common one being register-based prospective studies (*n* = 11) [23,24,25,26,27,28,29,30,31,32,33], and one was a case-control study [34]. Six of the twelve studies were published in 2018, two were published in 2019 and one each in the years 2010, 2012, 2015, and 2016. The size of the populations included in the studies varied from 3666 to 1,466,100 individuals. Most studies were performed in Finland [24,25,26,27,28,30,32], followed by Sweden [29,31,33,34], and one study from Norway [23]. No studies were found from Denmark and the Faroe Islands or Iceland. The participants in the studies were the general working population of a country, region, or city. In three studies the populations consisted of employees in social services, municipal services, or health care [27,30,32]. One study examined twins and parents of children/young adults aged 16–24 years [31]. One study examined gender differences in sickness absence with psychiatric disorders [29].

The measures of domestic factors found in the studies were marital status, family situation (residential status, cohabitation, having children), work-home interference (in both directions and total workload), social affiliation, and loss of a child/young adult (aged 18–25 years). Marital status appeared in nine of the twelve included studies and family situation in five studies while work-home interference, social affiliation, and loss of a child appeared in one study each. The most common type of data was national register data (Table 1). 

Studies most frequently used a sick leave period of >9 or >14 days, but longer periods of time such as >30 and ≥60 days, respectively, appeared in single studies. Studies that used the measures of domestic factors for adjusting for confounding, and where the isolated effect of the variable was not possible to identify, could not be included in the analysis of associations (research question iii). Detailed information on these eight studies can be found in Table 1. Four studies directly employed the variables of domestic factors as an exposure. These studies are presented with the main findings in Table 2.

### 3.2. Association between Exposure Measures of Domestic Factors and Sickness Absence

#### 3.2.1. Marital Status and Family Situation

Two studies examined the influence of marital status and family situation on sickness absence. Lidwall et al. (2018) examined sickness absence with medically certified psychiatric disorders in a working population using demographic, socioeconomic, and occupational predictors [29]. Exposure variables analyzed were sick leave history, sex, age, marital status (married, unmarried, divorced, widowed, unknown), children in the family and their age (no children, children aged 0–2, 3–8, 9–12, 13–15 years), immigrant status, labor income, disability pension, and finally, place of residence, education, employment sector, occupation, and occupational status. In adjusted models where the mentioned variables were used as co-variates, marital status was associated with a higher risk of sickness absence with psychiatric disorders compared to all-cause sickness absence. Associations were strongest for divorced people, but also widowed and unmarried people had a higher risk compared to married people. It was also found that parents of children older than two years had a higher incidence of sickness absence with psychiatric disorders. 

#### 3.2.2. Work-Home Interference

Svedberg and colleagues (2018) examined work-home interference to predict sickness absence (SA) with stress-related diagnoses or with other mental disorders [31]. Three exposure measures were used: home-to-work conflict, work-to-home conflict, and total workload. After adjusting for age, education, marital status, living with children, working full time, job demands, control, support, previous sick leave, and self-rated health, only two odds ratios remained significant. The adjusted results showed that women exposed to work-to-home conflict had a statistically significant higher odds ratio for sickness absence with other mental disorders compared to women who were not exposed. None of the three exposure measures showed significant odds ratios after adjustment for sickness absence with stress-related diagnoses. For men, the association between work-to-home conflict was significant in sickness absence with stress-related diagnosis after adjustment of co-variates but not for sickness absence with other mental disorders or for any of the two other exposure measures. 

#### 3.2.3. Social Affiliation

In a Norwegian register-based 5-year prospective cohort study by Foss et al. (2010), the researchers examined both work-related and individual factors as possible predictors of long-term sickness absence with psychiatric disorders defined as >8 weeks [23]. Several variables were analyzed, such as socioeconomic status, occupational factors, mental distress, self-reported health, social affiliation, work-related health, smoking, and alcohol use. Social affiliation was measured by two questions: “Do you feel you have enough good friends?” (Response categories: Yes or No) and “How often do you take part in some kind of club/social activity?” (Response categories: Never, A few times per year, and 1–2 times per month or more). Social affiliation was used as a potential determinant separately for men and women for a 5-year risk of long-term sickness absence with psychiatric disorders. The risks were slightly higher for having no good friends versus having good friends and for few social activities as compared to social activities >1–2 times per month. These results applied to both women and men. The findings were not highlighted by the authors, and we conclude that the effect of social affiliation on hazard ratios for long-term sickness absence with psychiatric disorders was found to be modest.

#### 3.2.4. Death of Offspring 

In a study by Wilcox and colleagues (2015), the researchers analyzed whether there was an association between parents who lost a child (aged 16–24 years) to suicide, an accident, or natural death, and sickness absence with psychiatric disorders [33]. Analyses were adjusted for age, marital status, level of education, country of birth, urbanization, health care due to mental illness, suicide attempt, previous sick leave, and previous early retirement in parents. Adjustment for child-related factors was performed for age, gender, previous outpatient and/or inpatient care due to mental illness, and suicide attempts. The researchers found that mothers and fathers who lost a child (aged 16–24 years) to suicide or an accident had a ten times higher risk of sickness absence with psychiatric disorders at follow-up compared to other parents. The risk was four to six times higher for a natural death.

## 4. Discussion

This scoping review could only identify twelve studies performed in any of the Nordic countries (Denmark, Finland, Iceland, Norway, and Sweden) during the period 2010–2019 that in one way or another studied domestic factors. The 12 studies were identified after a title and abstract screening of 1167 articles. Of the 12 studies, 4 had a measurement of a domestic factor as exposure. Only two of the twelve studies stated a specific purpose for studying the narrower concept of domestic or unpaid work. In relation to the unexplained differences in sickness absence with psychiatric disorders between women and men, this finding is discouraging, especially since the Nordic countries were included. Nordic countries are recurrently in the top of gender equality rankings [35], and it can be assumed that opportunities for research on domestic or unpaid work are good. The outcome of interest in this review, sickness absence, is highly researched in the Nordic countries, so the lack of studies is mainly related to the absence of research on domestic or unpaid work. A possible explanation for the few studies identified is the fact that studies on sickness absence are often based on public registers, which is an advantage with high coverage and quality. However, data on exposures are scarce in registers regarding details in relation to paid work and absence for domestic or unpaid work. Domestic factors such as demographic information on marital status and No. of children are available in registers, and we found in this review that marital status and family situation were the most common measures of domestic factors. Thus, the data availability guides what can be studied. Additionally, preferred scientific methods such as multivariate regression modeling reduce the possibility to identify the association between a single variable and the outcome. In eight studies, measures for domestic factors were included as confounders and contributed to answering our second research question on the type of measurements used. Another possible reason for few studies on domestic factors may be the connection in rules and regulations between sickness absence and paid work. However, health problems can occur through exposure to detrimental factors in the domestic sphere. Sustainable health must also integrate both perspectives in relation to sickness absence with psychiatric disorders. Globally, the distribution of responsibilities within the domestic sphere is highly gendered. Women and men have different responsibilities, and women most often use more time for domestic responsibilities than men do. Of the four studies that specifically measured domestic factors as exposure (Table 2), three [23,31,33] performed gender-stratified analyses. One study [23] did not find any association between the domestic factor studied. The two others [31,33] found associations but no gender differences in risk for sickness absence with psychiatric disorders in the adjusted models. This implies that women and men exposed to demands from domestic factors may be affected in similar ways. However, women still outnumber men in numbers exposed, which can explain the gender gap in sickness absence with psychiatric disorders even if this conclusion cannot be drawn based on the findings in this review. More studies are needed with better measurements of domestic factors to test this hypothesis further.

One of the more striking findings in this review was the very high increased risk of sickness absence with psychiatric disorders among parents who had lost a child aged 16–24 years to suicide, accidents, or natural causes. The emotional load of these types of events is of course very high and the finding as such is not surprising. It may be argued that the loss of a child is an event of a magnitude of its own and, therefore, is misplaced within the concept of domestic factors. We think that events with strong emotional load can be incorporated since they also include practical, juridical, and relational issues. Parents may be responsible for the grief of siblings, friends, and other relatives. Parents may be divorced, and information exchange is needed between the two sides. Thus, the loss of a child is traumatic and life changing but is also associated with several issues that justify its inclusion in domestic factors. The possibility of preventing sickness absence with psychiatric disorders could probably be improved if parents who experience loss of a child are supported. Future studies should look more closely into the patterns of temporal proximity between death and sickness absence, the association with health care visits and treatment with medication and/or psychotherapy, and parents’ occupations and/or socioeconomic status. Finally, in the case of siblings to the decedent, future studies should also incorporate them to get a full picture of how a death of a young person affects the family. 

The results from this review show that there still are major knowledge gaps in research on domestic factors and sickness absence with psychiatric disorders, in particular the narrower aspect of domestic or unpaid work. This is in sharp contrast to the extensive research on domestic factors and mental disorders [6,7,8,9]. The difference between married or cohabiting individuals and those divorced or widowed is well known, which contributes to the use of these factors as confounders in many of the studies included in this review. 

Domestic factors are a broad concept, and several different angles can be identified. There is a lack of research regarding sickness absence with psychiatric disorders, and we have compiled our knowledge to suggest that domestic factors can be approached from at least six different overarching angles in relation to sickness absence with psychiatric disorders: “children”, “time use”, “tasks”, “total workload”, “roles and relationships”, and “fair division of domestic influence”. Each of these angles can be divided into several aspects. A few aspects of particular importance in relation to sickness absence with psychiatric disorders are: The importance of *children* with special needs (e.g., functional variations, social problems);How different types of work *tasks* (e.g., practical tasks, planning tasks, relationship tasks) are divided between partners also influences the level of total workload;Traumatic life events and emotional strain in more acute and more prolonged situations (e.g., serious illness or death in close *relationships*);More severe psychosocial problems in *relationships* (e.g., domestic violence or abuse).

An important area for further exploration is understanding how these aspects may contribute to the risk of sickness absence due to psychiatric disorders. Method development is specifically needed to be able to expand the focus of the studies beyond, for example, the number of children, the children’s ages, and whether the children receive compensation for special needs. Studies of children are limited to parents with children at home, but women’s higher sickness absence is also present between women and men without children at home and who are not parents. Uneven distribution of home, household, and care work, relationship problems, same-sex relationships, and violence in close relationships are some sub-topics that need to be studied in more detail to get a more comprehensive picture of possible causal factors. In particular, the concept of “unpaid domestic work” needs theoretical and empirical development. 

A challenge in the future development of this research area is that domestic factors by definition are performed in the private sphere of life. There are ethical issues that need to be considered. In addition, domestic or unpaid work is often performed with and for people whom an individual has an emotional connection with, more so than paid work. Consequently, the demands of domestic or unpaid work can be both stressful and fulfilling to a degree that is not as commonly encountered in paid work. 

### Methodological Considerations 

An important methodological consideration for this review would be that it only considers studies in the Nordic region, limiting the number and nature of studies. On the other hand, the countries examined are known to rank high in gender equality indexes.

Studies were excluded when it was not possible to specifically read results for sickness absence with psychiatric disorders. Studies including sickness absence irrespective of diagnosis are common, but they include reasons for absence such as infectious diseases, chronic disorders, cancer, musculoskeletal disorders, and psychiatric disorders. For this study, we wanted the specific associations between domestic factors and sickness absence with psychiatric disorders since we assume that the added demands from domestic or unpaid work and other aspects of family life might be more pronounced in this group of disorders.

A strength of this review is that all 12 studies identified were prospective cohort studies based mainly on different administrative and public registers. Public registers of sickness benefits paid are of good quality and have few opportunities for error. The studies are also based on slightly longer sick leave, which means that there is a medical certificate from a doctor as a basis. This reduces the sources of error regarding the cause of sick leave, i.e., the type of illness or the type of symptoms on which sickness absence is based. As was foreseeable, controlled trials were absent; sickness absence is based on universal social insurance in the Nordics, to which the citizen is entitled if requirements are met.

## 5. Conclusions

From this review of Nordic research published in scientific journals during the years 2010–2019, we conclude that there are major knowledge gaps in research on domestic factors and sickness absence with psychiatric disorders. Demographic variables such as marital status and family situation dominated the measures used, probably related to their access in public registers. Though by account of the very few studies that have examined domestic factors and the limited scope of the existing research, more knowledge is needed in virtually all conceivable areas related to domestic factors. Future studies need to develop concepts, study designs, and measurements to move this research area further.

## Figures and Tables

**Figure 1 ijerph-20-06292-f001:**
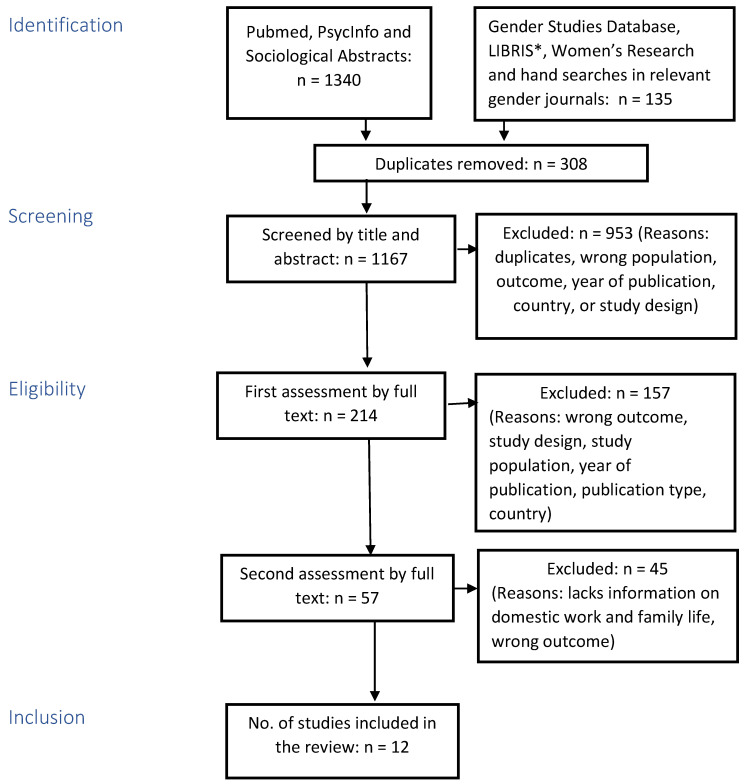
Flow diagram of the search strategy (*Swedish libraries search service).

**Table 1 ijerph-20-06292-t001:** Characteristics of included studies (*n* = 12) in the scoping review on domestic factors and sickness absence with psychiatric disorders, publication years 2010–2019.

Study	Author, Publishing Year, Country, Title	Study Design, Population	Main Exposure	All Confounders
**1**	Foss, L., et al. (2010), [23] Norway, *Risk factors for long-term absence due to psychiatric sickness: a register-based 5-year follow-up from the Oslo health study*	Register-based 5-year prospective cohort study, 8333 inhabitants in Oslo who were judged to be at risk of being sick with a mental illness	SEP, occupational factors, mental distress, self-reported health, work-related health, smoking and alcohol use, **social affiliation (social activities > 1–2 times/month, more seldom, missing)**	Age, level of education, work-related health, work-related factors, self-reported general and mental health, lifestyle, and **social affiliation**
**2**	Halonen, J. I., et al. (2018), [24] Finland, *Mental health by gender-specific occupational groups: profiles, risks, and dominance of predictors*	Register-based 9-year prospective cohort study, 414,357 randomly selected population cohort that included 33% of the 18–64-year-old permanent Finnish residents at the end of 2004	Gender-specific profiles for mental ill-health based on the 6 largest occupational groups for women and the 6 largest occupational groups for men, one of which was common (service profession)	Age, income, and unemployment in previous year **Marital status**
**3**	Harkko, J., et al. (2018), [25] Finland, *Unemployment and work disability due to common mental disorders among young adults: selection or causation?*	Register-based 5-year prospective cohort study, 116,829 individuals out of a larger sample of 119,061 which was 60% of the population in Finland in 2005	An unemployment benefit claim, unemployment	Educational level, residential area, and the number of years registered as not in employment or education, purchases of psychotropic medication, participation in psychiatric rehabilitation, and previous sickness allowances due to mental disorders, **residential status (parental home, single,** **own family, without permanent residence)**
**4**	Kaila-Kangas, L., et al. (2018), [26] Finland, *Alcohol use and sickness absence due to all causes and mental- or musculoskeletal disorders: a nationally representative* *study*	Register and survey-based 10-year prospective cohort study, 3666 individuals aged between 30 and 55 years out of a larger sample of 8028.	Alcohol use	Age, sex, education, CMD, chronic diseases, BMI, smoking, leisure time, job strain, occupational status, physically strenuous work **Marital status**
**5**	Kokkinen, L., et al. (2019), [27] Finland, *Human service work and long-term sickness absence due to mental disorders: a prospective study of gender-specific patterns in 1,466,100 employees*	Register-based 9-year prospective cohort study, 1,466,100 individuals aged between 25 and 55 years, which was 33% random sample of the working-age population (18–64 years at baseline) in two consecutive cohorts (1996–2004 and 2005–2013)	Occupation, human service occupations are divided into five categories: health care, education, social work, customer service and a mixed group of police officers and psychologists	Age, income, level of education, country of residence, and unemployment at baseline **Marital status**
**6**	Leinonen, T., et al. (2018), [28] Finland, *Labour Market Segregation and Gender Differences in Sickness Absence: Trends in 2005–2013 in Finland*	Register-based 9-year prospective cohort study, 1,097,598 employees with salary in Finland who were 25–59 years at baseline in 2005, 1,122,238 individuals in 2008 and 1,080,951 individuals in 2013. The study group came from a representative random sample of 70% of the population	Occupation by level of education: 3 categories from top officials to manual workers. Industry, 9 categories. Subdivision from Statistics Finland.	Education, income employment sector, time spent in employment, age **Marital status, having children**
**7**	Lidwall, U., (2016), [34] Sweden, *Effort–reward imbalance, overcommitment, and their associations with all-cause and metal disorder long-term sick leave—a case-control study of the Swedish working population*	Case-control study, 3477 individuals, 20–64 years in Sweden, with a long-term sick leave > 59 days from a larger cohort of 16,298 individuals. A control group with 2078 employees received from Statistics Sweden	Effort–reward imbalance (ERI), overcommitment (OC)	Age, education, employment status (permanent, fixed-term self-employment), occupation, category of employment (state, municipality, etc.), working time, physical work environment, smoking, obesity, and self-reported health **Cohabitation and minor children at home**
**8**	Lidwall, U., et al. (2018), [29] Sweden, *Mental disorder sick leave in Sweden: A population study.*	Register-based 1-year prospective cohort study, 6,192,397 persons out of all insured residents aged between 16 and 64 years as of 31 December 2011.	Sick leave history, sex, age, immigrant status, labor income, disability pension, and finally, place of residence, education, employment sector, occupation, and occupational status **Marital status, children in the family and their age**	Sick leave history, sex, age, immigrant status, labor income, disability pension, and finally, place of residence, education, employment sector, occupation and occupational status **Marital status, children in the family and their age**
**9**	Mauramo, E., et al. (2019), [30] Finland, *Changes in common mental disorders and diagnosis-specific sickness absence: a register-linkage follow-up study among Finnish municipal employees.*	Register-based 5-year prospective cohort study, 3890 employees of the city of Helsinki aged 40 to 60 years old. Survey data from the Helsinki Health Study collected in two phases (2000–2002 and 2007, respectively) were linked through personal identification numbers to register data.	Change in mental illness: no mental illness, favorable changes in mental illness, unfavorable changes in mental illness, and repeated periods of mental illness	Gender, age, occupation by level of education, overtime, shift work physical workload, psychosocial working conditions, current smoking, heavy drinking, and limiting long-term illness **Marital status**
**10**	Svedberg, P., et al. (2018), [31] Sweden, *Work-Home Interference, Perceived Total Workload, and the Risk of Future Sickness Absence Due to Stress-Related Mental Diagnoses Among Women and Men: a Prospective Twin* *Study*	Register-based prospective cohort study with twins, 7–9 years follow-up (baseline June 2004 with follow-up 2013), 11,916 twins, 19–47 years old in Sweden in 2005, based on a larger cohort of 25,496 twins born in 1959 and 1985	**Work-to-home conflict, home-to-work conflict, total workload**	Age, sex, work status, job demand, control, support, self-rated health, previous sick leave **Marital status, living with children**
**11**	Virtanen, M., et al. (2012), [32] Finland, *Health risk behaviours and morbidity among hospital staff—comparison across hospital ward medical specialties in a* *study of 21 Finnish hospitals*.	Register and survey-based 1-year prospective cohort study, 8003 medical staff members out of 13,229 from 21 hospitals in 6 Finnish hospital districts.	Medical speciality, lifestyle, diagnosed diseases	Age, gender, occupation (physicians/other professionals, nurses, practical/ assistant nurses), type of employment (permanent/ temporary), hospital district, ward type (inpatient bed ward versus outpatient clinic), and work unit medical specialty, **Marital status**
**12**	Wilcox et al. (2015), [33] Sweden, *Functional impairment due to bereavement after the death of adolescent or young adult offspring in a national population study of 1,051,515 parents*	Register-based prospective cohort study with 1–3 years follow-up (baseline 2004, follow-up July 2005), 1,051,515 mothers and fathers with children/young adults aged 16–24 years in Sweden 31 December 2004	**Loss of child/young adult (cause of death: suicide, accident, or natural death)**	Parents: age, level of education, country of birth, urbanization, health care due to mental illness, suicide attempt, previous sick leave, and previous early retirement. Child/young adult: age, gender, former open-label and inpatient care due to mental illness and attempted suicide **Marital status**

Measures of domestic factors are marked with bold text to facilitate identification when reading the table. SEP is short for SocioEconomic Position. CMD is short for Common Mental Disorders. BMI is short for Body Mass Index.

**Table 2 ijerph-20-06292-t002:** Characteristics of and findings in studies that used measures of domestic factors as exposure, publication years 2010–2019 (results from adjusted analyses).

Measures of Domestic Factors	Author, Year of Publication, Country	Findings
Marital Status	Lidwall, U., et al., 2018, Sweden [29]	Being divorced, unmarried, and widowed were associated with sickness absence with psychiatric disorders.
Family situation		Parents of children older than two years had a higher incidence sickness absence with psychiatric disorders.
Work-to-home conflict	Svedberg, P., et al, 2018, Sweden [31]	Work-to-home conflict was associated with sickness absence with stress related diagnoses in men and with sickness absence with other mental disorders in women.
Social Affiliation	Foss, L., et al., 2010, Norway [23]	Social affiliation was not associated with long-term sickness absence with psychiatric disorders.
Loss of child/young adult (cause of death: suicide, accident, or natural death)	Wilcox et al. 2015, Sweden [33]	Parents of suicide, accident, and natural decedents were found to be at higher risk for sickness absence with psychiatric disorders than parents who had not experienced this emotional load.

## Data Availability

No new data were created or analyzed in this study. Data sharing is not applicable to this article.

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
