# Peer review of "Domestic Factors as Determinant of Sickness Absence with Psychiatric Disorders: A Scoping Review of Nordic Research Published between 2010–2019"

_ijerph, 2023, doi:10.3390/ijerph20136292_

Round 1

Reviewer 1 Report

Thank you for the opportunity to review this manuscript.

This review seeks to fill an important gap in the extant research in examining factors outside of paid work that may contribute sickness absence due to psychiatric disorders. Firstly, my congratulations to the authors on their work to date, and for their pursuit of further knowledge in the area of domestic work and domestic determinants given they have been long overlooked as determinants of mental health. However, I do have a few issues regarding the review as it currently stands.  My major issue is outlined below, followed by some minor comments.

Major issues

I have a major issue with using the terminology “domestic work” to describe the determinant being explored in this review. “Domestic work” implies unpaid work to maintain a household.  Yet, the measures selected as indicators of “domestic work” do not actually fit this term. For example, death of a child – this is not an indicator of unpaid or domestic work.  Yes, it will cause enormous stress and grief, and so yes it will be associated with mental health (MH) and psychiatric disorders, but not through domestic/unpaid work pathways. Likewise for social affiliation and marital status (divorced, unmarried and widowed status) …. I do not think these are valid proxy measures for domestic work burden to be used as indicators of domestic work in this review. 

As per line 45 in your manuscript, “The unexplained part raises the question if non-work-related factors play a role in sickness absence with psychiatric disorders.” So, if the gap is in exploring the non-work-related factors that play a part, it is my strongest recommendation that the paper title and all associated terminology regarding the “domestic determinant” should instead be named something like “home factors” or “domestic factors” or “family life factors” or equivalent… rather than domestic work. This will still support your identified gap/rationale but most importantly it will align/make sense with respect to the domestic indicators that have been identified.

Please note, that in line with this expanded/alternative terminology, the background and justification for the study will need revision, as none of the 12 included studies pertain to unpaid labour specifically and yet it is unpaid labour and its association with MH that are introduced at the start of the paper.  I believe that the authors need to refocus the introduction onto family life factors that influence MH (including domestic labour) rather than be focused solely on unpaid/domestic labour/ work.  

The discussion section will also require rethinking considering the above.  For example, at present, there is no discussion of prior evidence (of which there is plenty) for some of the important family life factors (e.g., marital status, social connection/loneliness and household structure) included in the review as domestic determinants and their impact on MH. 

I appreciate that the initial intention of this review may have been to study domestic labour but that no studies on unpaid work and absenteeism due to mental illness were found/available.  If this is the case, then I think this is important to be discussed and highlighted as a gap – but nonetheless the resultant review that has emerged from the available literature (now focused on family life/domestic factors) needs to be titled, introduced and discussed appropriately.

Minor issues

1)     Line 44 “The gender distribution into male vs female dominated work sectors might influence sickness absence due to differential exposures.”   Differential exposures to what? Can you please expand what you mean here? 

2)     Figure 1 – 308 duplicates removed in box 3 – but duplicate removal also stated as one of reasons for exclusion in box 5 (n=953) – is this an error – had they not already been removed?

3)     Line 97 “(One was the first author GH. The second was PhD Maria Boström who has quit working and is not a co-author.)” I was wondering why MB would not still be included as a co-author if they conducted all the screening - even if no longer working at the Uni?

4)     Line 110 “When faced with disagreements on narrowing down the themes, we discussed our reasoning and reached a consensus.”  Suggest including initials of who charted date and decided on themes (as you have done for screening).

5)     Line 119 “All included studies had an epidemiological study design; the most common one being register-based prospective studies (n=11), and one was a case-control study [23-34].”  Suggest citing the 11 studies after n=11 and the remaining study after CC study… so readers know which are which.

6)     Line 121 “Six of the 12 studies were published in 2018.”  What about the other 6? 

7)     Line 122 “Most studies were performed in Finland, followed by Sweden and one study from Norway.” Suggest insert citations/references for each of these subgroups.

 8)     Line 125 “In four studies the populations consisted of employees in social services, municipal services, or health care.” As per comment 7 – suggest citing the 4 studies this statement refers to and suggest do the same for all at subsequent descriptive results sentences. 

9)     Section 3.2 – Regarding all descriptions of the associations reported in included studies: I would suggest that (unless you have reason to question the confounding adjustment of included studies), readability, flow and relevance would be enhanced if you only summarise the findings from adjusted models (and omit crude/unadjusted results for the purposes of the review).

The quality of the English is currently sufficient and allows the reader to make sense of the study. However, I believe that editing by a proficient English editor (for minor-moderate grammatical and sentence structure issues) will greatly enhance the readability and quality of the study (when the review reaches publication stages).

Author Response

Reviewer 1

Thank you for relevant and helpful suggestions.

Major issues

The terminology

We agree that the term domestic work is not the most appropriate and have changed to your suggestion “domestic factors”. That expression is broader and more relevant. In some parts of the manuscript, we use “domestic or unpaid work” when we specifically refer to work needed to maintain a household.  

Introduction and discussion

The introduction is based on studies using “domestic work” as a collective term for different aspects of exposures in the domestic sphere. Different aspects, other than household maintenance, are included in the studies for example age, gender and marital status. In line with your objection above, probably some of these studies also should have used “domestic factors”. After changing term from domestic work to domestic factors we think that the introduction works better as a starting point for the review. It also mirrors our a priori assumptions. By doing the review we gained new insights. We are of course willing to re-write the introduction if considered necessary.

Regarding the discussion, we have added a few lines on domestic determinants and their impact on mental health. Since the focus is on sickness absence and psychiatric disorders we have strived to avoid going to deep into research on mental health in general. One reason for a narrower focus is that sickness absence with psychiatric diagnose has increased considerably the last decade in the Nordic countries, and there is a need to understand mental health in the context of reduced work capacity.

Minor issues

1)     Line 44 “The gender distribution into male vs female dominated work sectors might influence sickness absence due to differential exposures.”   Differential exposures to what? Can you please expand what you mean here? 

This has been clarified to …differential work-related exposures.

2)     Figure 1 – 308 duplicates removed in box 3 – but duplicate removal also stated as one of reasons for exclusion in box 5 (n=953) – is this an error – had they not already been removed?

There was an initial removal of duplicates made by the librarians. Then, in the step where we screened titles and abstracts a few more duplicates were found.

3)     Line 97 “(One was the first author GH. The second was PhD Maria Boström who has quit working and is not a co-author.)” I was wondering why MB would not still be included as a co-author if they conducted all the screening - even if no longer working at the Uni?

We have clarified that Maria Boström was asked to participate but declined since she is no longer working with research. So, it was her own choice to not participate. (She is now working with textile crafts).

4)     Line 110 “When faced with disagreements on narrowing down the themes, we discussed our reasoning and reached a consensus.”  Suggest including initials of who charted date and decided on themes (as you have done for screening).

We have added initials.

5)     Line 119 “All included studies had an epidemiological study design; the most common one being register-based prospective studies (n=11), and one was a case-control study [23-34].”  Suggest citing the 11 studies after n=11 and the remaining study after CC study… so readers know which are which.

Details on references have been added.

6)     Line 121 “Six of the 12 studies were published in 2018.”  What about the other 6? 

Details on the other six have been added.

7)     Line 122 “Most studies were performed in Finland, followed by Sweden and one study from Norway.” Suggest insert citations/references for each of these subgroups.

References for each subgroup have been added.

8)     Line 125 “In four studies the populations consisted of employees in social services, municipal services, or health care.” As per comment 7 – suggest citing the 4 studies this statement refers to and suggest do the same for all at subsequent descriptive results sentences. 

Thank you for pointing at this. In fact, the number should be 3 studies and the references for these three studies have been added.

9)     Section 3.2 – Regarding all descriptions of the associations reported in included studies: I would suggest that (unless you have reason to question the confounding adjustment of included studies), readability, flow and relevance would be enhanced if you only summarise the findings from adjusted models (and omit crude/unadjusted results for the purposes of the review).

We have removed descriptions of unadjusted results.

Reviewer 2 Report

I think that the article is well worked and well done, the research questions are clear and the conclusions are well understood. The authors explain their conclusions adequately on a highly relevant and little-studied topic. Despite the fact that they comment on the design of the studies, I would like to ask them if they have not considered incorporating quality quantitative data such as the Jadad scale and if they have not considered the option of performing a meta-analysis. Perhaps they could reflect on these aspects in the discussion, of course if they see it necessary.

Author Response

Thank you for your encouraging words on the paper.

We agree that the Jadad scale or similar quality assessment tools are valuable. However, in a scoping review quality assessments are not necessarily included. In this study the main objective was to get an overview of the research field and specifically to scrutinize the type of measurements used. The results from different studies were also important but given the few studies identified, and the varied measurements used, a meta-analysis was not seen as meaningful. The inclusion of quality assessments scales for quantitative data, such as the Jadad scale, can provide additional insights into the rigor of the studies, we think it should be considered in future research on this topic. We have pointed to the quality aspects of included studies (register studies, large populations, prospective design). However, we did not add anything on quality assessments scale since we choose a scoping review design.

Reviewer 3 Report

The article compiles current Nordic research on domestic work and sickness absence with psychiatric disorders. The paper is fascinating and worthy of publishing. I have only two minor comments.

-Authors could emphasize in the beginning of the introduction why this study is important. 

- Why did authors select the time period from 2010 to 2019? Why are the newest studies leaved out ?

Author Response

The article compiles current Nordic research on domestic work and sickness absence with psychiatric disorders. The paper is fascinating and worthy of publishing. I have only two minor comments.

Thank you for the kind words.

-Authors could emphasize in the beginning of the introduction why this study is important. 

Thank you for the suggestion. We have added the following at the end of the first paragraph:

With this scoping review we add to existing knowledge on domestic factors and their impact on mental health. In today’s society, sustainable health is likely to be achieved if both factors in paid work and domestic factors are better understood as detrimental to and promotive of health.  

- Why did authors select the time period from 2010 to 2019? Why are the newest studies leaved out ?

The time period was set at start of the project which was in 2020. At that point the last year possible to search was 2019.Some reasons for the delay of submitting the manuscript relate to authors’ different obligations in academia which led to restricted time for research. 

Round 2

Reviewer 1 Report

Thank you for considering the review comments and making suggested changes.  The paper is much improved now that the intended exposure has been expanded to domestic factors (rather than restricted to domestic work).

Given this has been addressed, I now have a couple of additional comments.

1) I would like to see a couple of sentences/short paragraph added to the discussion that discusses any gender differences revealed by your review (for the included studies that stratified their results by gender) ... especially given exposure to domestic factors is such a gendered experience in most parts of the world.  Were Nordic women more vulnerable to sickness absence due to domestic factors compared to men? - ie. what was your overall take from the existing evidence?  - especially given the nuance of the start of your discussion regarding the nordic gender equality ratings - link your findings to this? discuss nordic gender norms etc. 

2) I am still somewhat struggling to see how the loss of a child constitutes a domestic factor in the way you are linking such factors to unpaid/domestic work throughout your manuscript (as it does not really tie into unpaid work in my view).  Therefore, I would like to see some brief explanation/rationale in the paper as to why this particular exposure was included as a domestic factor. 

The following makes sense ...

By discussion we have tried to compile our knowledge formed by these sources to suggest that domestic work can be approached from at least six different overarching angles in relation to sickness absence with psychiatric disorders: “children”, “time use”, “tasks”, “total workload”, “roles and relationships”, and “fair division of domestic influence”.

But not (so much) the subsequent way these angles have then been divided into aspects - one of which is "traumatic life events" ... I suggest you need some stronger conceptual framework to tie this aspect in to your narrative - can you cite some pre-existing literature that ties the death of a loved one/child in as a domestic factor/related to unpaid work?  Conceptually, having a sick/ ill loved one increases care requirements/unpaid work and so the link to sickness absence is evident with family illness under the framework of unpaid time requirements - but the death of child has very different pathways to sickness absence.  Please either provide a rationale/pathway for this under the umbrella of domestic work (which is where most of the theoretical underpinning of this paper lies), or add some more detail in the introduction regarding different/other types of domestic factors unrelated to unpaid work (such as death of a child) as you see them and their justification. 

Well done on an important piece of work. 

Minor editing in the proofing process will help with readability and flow as some of the grammar could be tightened (but hats off to the authors for being able to write a manuscript in English so well if it is not their first language).

Author Response

Reviewer 1

Gender differences

This is an interesting point and we have added the following paragraph:

Globally, the distribution of responsibilities within the domestic sphere is highly gendered. Women and men have different responsibilities, and women most often use more time for domestic responsibilities than men do. Of the four studies that specifically measured domestic factors as exposure (Table 2), three [23,32, 34] performed gender stratified analyses. One study [23] did not find any association between the domestic factor studied The two other [32, 34] found associations but no gender differences in risk for sickness absence with psychiatric disorders in the adjusted models. This imply that women and men exposed to demands from domestic factors may be affected in similar ways. However, women still outnumber men in numbers exposed which can explain the gender gap in sickness absence with psychiatric disorders even if this conclusion cannot be drawn based on the findings in this review. More studies are needed with better measurements of domestic factors to test this hypothesis further.

Loss of a child

We agree that the loss of a child less intuitively incorporated in the concept of domestic factors. However, we think that the situation of losing a child apart from the trauma and emotional load also mean several practical, juridical, economic and relational issues. We have developed this line of argumentation in the discussion.

It may be argued that the loss of a child is an event of a magnitude of its own, and therefore is misplaced within the concept of domestic factors. We think that events with strong emotional load can be incorporated since they also include practical, juridical, and relational issues. Parents may be responsible for siblings’, friends’, and other relatives’ grief. Parents may be divorced, and information exchange are needed between the two sides. Thus, loss of a child is traumatic and life changing but also associated with several issues that justifies its inclusion in domestic factors.

Well done on an important piece of work

Thanks!